# On Scrambling Phenomena
# for Randomly Initialized Recurrent Networks

**Vaggos Chatziafratis**[*]
Department of Computer Science and Engineering
University of California, Santa Cruz
`vaggos@ucsc.edu`

**Ioannis Panageas**
Department of Computer Science
University of California, Irvine
`ipanagea@ics.uci.edu`

**Clayton Sanford**
Department of Computer Science
Columbia University
`clayton@cs.columbia.edu`

**Stelios Andrew Stavroulakis**
Department of Computer Science
University of California, Irvine
`sstavrou@uci.edu`

## Abstract

Recurrent Neural Networks (RNNs) frequently exhibit complicated dynamics, and their sensitivity to the initialization process often renders them notoriously hard to train. Recent works have shed light on such phenomena analyzing when exploding or vanishing gradients may occur, either of which is detrimental for training dynamics. In this paper, we point to a formal connection between RNNs and chaotic dynamical systems and prove a qualitatively stronger phenomenon about RNNs than what exploding gradients seem to suggest. Our main result proves that under standard initialization (e.g., He, Xavier etc.), RNNs will exhibit *Li-Yorke chaos* with *constant* probability *independent* of the network's width. This explains the experimentally observed phenomenon of *scrambling*, under which trajectories of nearby points may appear to be arbitrarily close during some timesteps, yet will be far away in future timesteps. In stark contrast to their feedforward counterparts, we show that chaotic behavior in RNNs is preserved under small perturbations and that their expressive power remains exponential in the number of feedback iterations. Our technical arguments rely on viewing RNNs as random walks under non-linear activations, and studying the existence of certain types of higher-order fixed points called *periodic points* that lead to phase transitions from order to chaos.

## 1 Introduction

In standard feedforward neural networks (FNNs), computation is performed "from left to right" propagating the input through the hidden units to the output. In contrast, recurrent neural networks (RNNs) form a feedback loop, transfering information from their output back to their input (e.g., LSTMs (Hochreiter and Schmidhuber, 1997), GRUs (Cho et al., 2014)). This feedback loop allows RNNs to share parameters across time because the weights and biases of each iteration are identical. As a result, RNNs can capture long range temporal correlations in the input. For these reasons, they have been very successful in applications in sequence learning domains, such as speech recognition, natural language processing, video understanding, and time-series prediction (Bahdanau et al., 2014; Cho et al., 2014; Chung et al., 2014).

---

[*]Authors order determined by the output of a randomly initialized recurrent network (operating at the chaotic regime).

36th Conference on Neural Information Processing Systems (NeurIPS 2022).

Unfortunately, their unique ability to share parameters across time comes at a cost: RNNs are sensitive to their initialization processes, which makes them extremely difficult to train and causes complicated evaluation and training dynamics (Le et al., 2015; Laurent and von Brecht, 2016; Miller and Hardt, 2018). Roughly speaking, because the hidden units of an RNN are applied to the input over and over again, the final output can quickly explode or vanish, depending on whether its Jacobian's spectral norm is greater or smaller than one respectively. Similar issues also arise during backpropagation that hinder the learning process (Allen-Zhu et al., 2018).

Besides the hurdles with their implementation, recurrent architectures pose significant theoretical challenges. Several basic questions include how to properly *initialize* RNNs, what is their *expressivity* power (also known as representation capabilities), and *why* do they converge or diverge, all of which require further investigations. In this paper, we take a closer look at randomly initialized RNNs:

*Can we get a better understanding of the behavior of RNNs at initialization using dynamical systems?*

We draw on the extensive dynamical systems literature—which has long asked similar questions about the topological behavior of iterated compositions of functions—to study the properties of RNNs with standard random initializations. We prove that under common initialization strategies, e.g., He or Xavier (He et al., 2015, 2016), RNNs can produce dynamics that are characterized by chaos, even in innocuous settings and even in the absence of external input. Most importantly, chaos arises with *constant* probability which is *independent* of the network's width. Our theoretical findings explain empirically observed behavior of RNNs from prior works, and are also validated in our experiments.[1]

More broadly, our work builds on recent works that aim at understanding neural networks through the lens of dynamical systems; for example, Chatziafratis et al. (2020b,a) use Sharkovsky's theorem from discrete dynamical systems to provide depth-width tradeoffs for the representation capabilities of neural networks, and Sanford and Chatziafratis (2022) further give more fine-grained lower bounds based on the notion of "chaotic itineraries".

## 1.1 Two Motivating Behaviors of RNNs

Before stating our main result, we illustrate two concrete behaviors of RNNs that inspired our work. The first example demonstrates that randomly initialized RNNs can lead to what is perhaps most commonly perceived as "chaos", while the second example demonstrates a qualitatively different behavior of RNNs compared to FNNs. Our main result unifies the conclusions drawn from these two.

**Scrambling Trajectories at Initialization** Prior works have empirically demonstrated that RNNs can behave chaotically when their weights and biases are chosen according to a certain scheme. For example, Laurent and von Brecht (2016) consider a simple 4-dimensional RNN with specific parameters in the absence of input data. They plot the trajectories of two nearby points $x, y$ with $\|x - y\| \leq 10^{-7}$ as they are propagated through many iterations of the RNN. They observe that the long-term behavior (e.g., after 200 iterations) of the trajectories is highly sensitive to the initial states, because distances may become small, then large again and so on. We ask:

*Are RNNs (provably) chaotic even under standard heuristics for random initialization?*

We answer this question in the affirmative both experimentally and theoretically. This question is helpful to understand for multiple reasons. First, it informs us about the behavior of *most* RNNs, as we start with a random setting of the parameters. Second, proving that a system is chaotic is qualitatively much stronger than simply knowing its gradient to be exploding; this will become evident below, where we describe the phenomenon of *scrambling* from dynamical systems. Finally, understanding why and how often an RNN is chaotic can lead to even better methods for initialization.

To begin, we empirically verify the above statement by examining randomly initialized RNNs. Figure 1 demonstrates that trajectories of different points may be close together during some timesteps and far apart during future timesteps, or vice versa. This phenomenon, which will be rigorously established in later sections, is called *scrambling* (Li and Yorke, 1975) and emerges as a direct consequence of the existence of higher order fixed points (called periodic points) of the continuous map defined by the random RNN.

---

[1]Our code is made publicly available here: `https://github.com/steliostavroulakis/Chaos_RNNs/blob/main/Depth_2_RNNs_and_Chaos_Period_3_Probability_RNNs.ipynb`

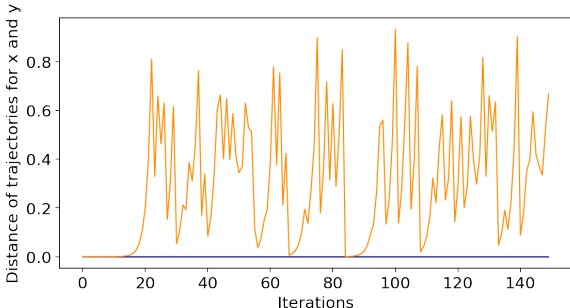

Figure 1: Points $x, y$ with initial distance $\|x - y\| \leq 10^{-7}$ and their subsequent $\|f^t(x) - f^t(y)\|$ distances across $t = 150$ iterations of a randomly initialized RNN. The idea behind scrambling is that the trajectories, even though they get arbitrarily close, they also separate later and vice versa.

**Persistence of Chaos in RNNs vs FNNs** Our second example is related to the expressive power of neural networks. A standard measure used to capture their representation capabilities is the number of linear regions formed by the non-linear activations (Montufar et al., 2014); the higher this number, the more expressive the network is. Counting the maximum possible number of linear regions across different architectures has also been leveraged in order to obtain depth vs width tradeoffs in many works (Telgarsky, 2015; Eldan and Shamir, 2016; Chatziafratis et al., 2020b,a).

Here, we are inspired by a remarkable observation of Hanin and Rolnick (2019a), who showed that real-valued FNNs from $\mathbb{R} \to \mathbb{R}$ may lose their expressive power if their weights are randomly perturbed or those weights are initialized according to standard methods. Roughly speaking, they show that in such networks the number of linear regions grows only linearly in the total number of neurons. These results contrast with the aforementioned analyses of the theoretical maximum number of regions, which is exponential in depth. We ask the analogous question for RNNs instead of FNNs:

*Will randomly initialized or perturbed RNNs lose their high expressivity like FNNs did?*

We show a contrast between RNNs and FNNs, by showing that the number of linear regions in random RNNs remains exponential on the number of its feedback iterations. Once again, the fundamental difference lies on the fact that RNNs share parameters across time. Indeed, the analyses of many prior works (Hanin and Rolnick, 2019a,b; Hanin et al., 2021) crucially relies on the "fresh" randomness injected at every layer, e.g., that the weights/biases of each neuron are initialized *independently* of each other; obviously, this is no longer true in RNNs where the same units are repeatedly used across different iterations. This shared randomness raises new technical challenges for bounding the number of linear regions, but we manage to indirectly bound them by studying fixed points of random RNNs.

## 2   Our Main Results

Our main contribution is to prove that randomly initialized RNNs can exhibit *Li-Yorke chaos* (Li and Yorke, 1975) (see definitions below), and to quantify when and how often this type of chaos appears as we vary the variance of the weights chosen by the random initialization. To do so, we use discrete dynamical systems which naturally capture the behavior of RNNs: simply start with some shallow NN implementing a continuous map $f$, and after $t$ feedback iterations of the RNN, its output will be exactly $f^t$ ($f$ composed with itself $t$ times). We begin with some basic definitions:

**Definition 2.1.** (Scrambled Set) Let $(X, d)$ be a compact metric space and let $f : X \to X$ be a continuous map. Two points $x, y \in X$ are called *proximal* if:

$$\liminf_{n \to \infty} d(f^n(x), f^n(y)) = 0$$

and are called *asymptotic* if

$$\limsup_{n \to \infty} d(f^n(x), f^n(y)) = 0.$$

A set $Y \subseteq X$ is called *scrambled* if $\forall x, y \in Y, x \neq y$, the points $x, y$ are proximal, but *not* asymptotic.

**Definition 2.2.** (Li-Yorke Chaos) The dynamical system $(X, f)$ is *Li-Yorke chaotic* if there is an uncountable scrambled set $Y \subseteq X$.

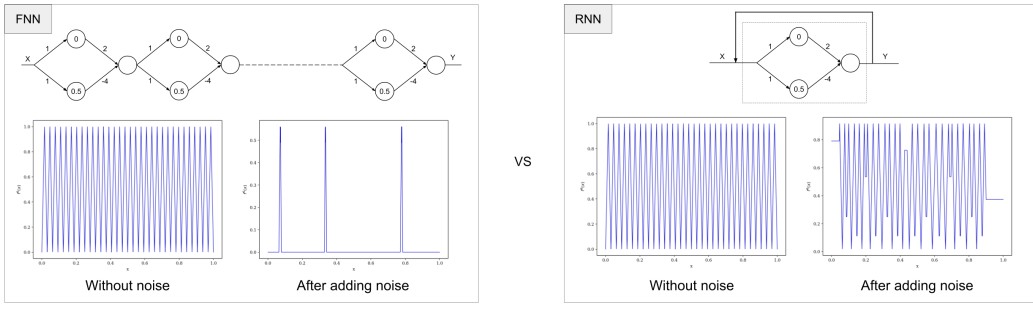

Figure 2: **Left:** Recreation of a figure from Hanin and Rolnick (2019a) where small Gaussian perturbation ($\mathcal{N}(0, 0.1)$) is added *independently* on every weight/bias of each layer of a simple FNN. As a result, after adding noise, the high expressivity (i.e., number of linear regions) breaks down; having "fresh" noise in each layer was crucial. **Right:** Here we depict the same network as before but shown as RNN, and we add noise as before. Perhaps surprisingly, RNNs exhibit different behavior than FNNs: high expressivity is preserved even after noise. As we show, due to *shared* randomness across iterations, small perturbations do not "'break" expressivity.

Li-Yorke chaos leads both to scrambling phenomena (Figure 1) and to high expressivity (Right of Figure 2). Interestingly, this is not true for FNNs (Left of Figure 2), where the fresh randomness at each neuron leads to concentration of the Jacobian's norm around 1, which is sufficient to avoid chaos. In other words, these definitions capture exactly the fact that trajectories get arbitrarily close, but also move apart infinitely often (as in Fig. 1). Intuitively, when scrambling occurs in RNNs, their #linear regions will be large and their input-output Jacobian will have spectral norm larger than one.

**Definition 2.3** (Simple RNN Model). For $k, \sigma > 0$, let $\mathcal{RNN}(k, \sigma^2)$ be a family of recurrent neural networks with the following properties:

- Input: The input to the network is 1-dimensional, i.e., a single number $x \in [0, 1]$.

- Hidden Layer: There is only one hidden layer of width at least $k$, with $\mathrm{ReLU}(x) = \max(x, 0)$ activations neurons, each of which has its own bias terms $b_i \sim \mathrm{Unif}([0, 1])$.

- Output: The output is a real number in $[0, 1]$ which takes the functional form $f_k(x) = \mathrm{clip}(\sum_{i=1}^{k} a_i \mathrm{ReLU}(x - b_i))$, where the weights are i.i.d. Gaussians $a_i \sim \mathcal{N}(0, \frac{\sigma^2}{k})$, and $\mathrm{clip}(\cdot)$ ensures that $f_k(x) \in [0, 1]$ similarly to the input (i.e., it "clips" the input so that it remains in $[0, 1]$ as it is an RNN).

- Feedback Loop: $f_k(x)$ becomes the new input number, so after $t$ iterations the output is $(f_k \circ f_k \circ \ldots \circ f_k)(x)$, i.e., $t$ compositions of $f_k$ with itself.

As we show, even this innocuous class of RNNs (see Sec. 3 for general model) leads to scrambling.

**Theorem 2.4** (Li-Yorke Chaos at Initialization). *Consider $f_k \in \mathcal{RNN}(k_\sigma, \sigma^2)$ initialized according to the He normal initialization (set $\sigma^2 = 2$, so weight variance is $2/k$). Then, there exists some constant $\delta > 0$ (independent of the width) and width $k_{He} > 1$, such that for sufficiently large $k > k_{He}$, $f_k$ is Li-Yorke chaotic with probability at least $\delta$.*

This answers our two questions posed earlier: RNNs may remain chaotic under initialization heuristics, and maintain their high expressivity, because Li-Yorke chaos implies an exponential #linear regions (Theorem 1.5 in Chatziafratis et al. (2020b)). Next, we focus on threshold phenomena:

**Theorem 2.5.** *[Order-to-Chaos Transition] For $f_k \in \mathcal{RNN}(k, \sigma^2)$, we get the following 3 regimes as we vary the variance of the weights $a_i \sim \mathcal{N}(0, \frac{\sigma^2}{k})$:*

- *(Low variance, order) Let $a_i \sim \mathcal{N}(0, \frac{1}{4k \log k})$. Then, the probability that $f_k$ is Li-Yorke chaotic is at most $\frac{1}{k}$.*

- *(Edge of chaos) Let $a_i \sim \mathcal{N}(0, \frac{2}{k})$ (i.e., He initialization). Then, $f_k$ is Li-Yorke chaotic with constant (albeit small) probability.*

- *(High-variance, chaos) Let $a_i \sim \mathcal{N}(0, \frac{\sigma^2}{k})$ for $\sigma = \omega(1)$. Then, $\lim_{k \to \infty} \Pr[f_k$ is Li-Yorke chaotic$] = 1$.*

To the best of our knowledge, we are the first to study Li-Yorke chaos in the context of RNNs and to prove that scrambling phenomena appear with constant probability independent of the width. We emphasize that Li-Yorke chaos is stronger than exploding gradients, as any chaotic system requires the input-output map to be non-contracting (as otherwise it would converge to a fixed point).

**Techniques**   Contrary to many prior works (Laurent and von Brecht, 2016; Miller and Hardt, 2018), we do not require differentiability of activations like $\tanh(\cdot)$, as we rely on properties of continuous functions. Most importantly, we deviate from past works (Bertschinger and Natschläger, 2004; Poole et al., 2016), as we do not use tools from mean-field analysis (where the number of neurons, width or depth goes to infinity), since here we are interested on the dependence on the width $k$. Instead, we study the fixed points and higher-order fixed points of $f_k \in \mathcal{RNN}(k, \sigma^2)$ (see Sec. 3), bounding the probability that $f_k$ has a certain type of fixed point known to lead to chaos.

This is done by viewing RNNs as random walks under non-linear activations. To give some intuition behind our findings, consider a random walk starting from $0$ in which we consider increments $d_i \sim \mathcal{N}(0, 1)$. It is quite well-known in the literature of random walks, that the expected number of steps needed for a random walk starting from $0$ with increments $d_i \sim \mathcal{N}(0, 1)$ to reach $n$ and come back to $0$ is $\Theta(n^2)$ iterates. For an RNN to be chaotic, as it turns out, we need to find the probability that it has certain type of fixed points; this can be bounded below by the probability that the random walk reaches point $1$ and then goes back to $0$. As a result, if we rescale appropriately and the increments $d_i \sim \mathcal{N}(0, \frac{1}{k})$, then we need $\Theta(k)$ iterates. Roughly speaking, $k$ plays the role of the size of the width ($d_i$ are the weights). In reality, the random walk we analyze is more complicated as the (random) biases affect the length of each increment in the random walk.

**Experiments**   Finally, we validate our theory by extensive simulations on a variety of randomly initialized RNN architectures. Our goal is twofold: Firstly, to demonstrate that Li-Yorke chaos is indeed present across different models and different initialization heuristics, and not just an artifact of the specificities of our $\mathcal{RNN}$ family. Secondly, to estimate the value of the aforementioned constant probability of chaos $\delta_\sigma$ in the Edge of Chaos regime, which is known to be the most interesting (Bertschinger and Natschläger, 2004; Yang and Schoenholz, 2017; Hanin, 2018), yielding a comparison for which initializations are more likely to be Li-Yorke chaotic (see table in Fig. 3).

## 2.1   Further Related Works

Several works have studied RNNs in the context of chaotic dynamical systems and edge of chaos initialization (Sompolinsky et al., 1988; Bertschinger and Natschläger, 2004; Saxe et al., 2013; Kadmon and Sompolinsky, 2015; Poole et al., 2016). They study neural networks in the limit of infinitely large widths or depth and derive properties of the dynamics, typically using mean-field analysis. Moreover, the notion of chaos used there is not from Li and Yorke (1975), as they do not analyze periodic points in the trajectories of the networks, nor do they establish scrambling phenomena. For example, Bertschinger and Natschläger (2004) observe that the sequence of activations in the neurons of the network exhibit increasingly unpredictable and complicated patterns as the variance increases and use this notion to determine the edge of chaos and threshold phenomena.

## 3   Preliminaries

**Higher-order Fixed Points**   Let $f : [a, b] \to [a, b]$ be a continuous map. The notion of a *periodic* point is a generalization of a *fixed* point:

**Definition 3.1.** We say $f$ contains period $n$ or has a point of period $n \geq 1$, if there exists a point $x_0 \in [a, b]$ such that:

$$f^n(x_0) = x_0 \quad \text{and} \quad f^t(x_0) \neq x_0, \forall \, 1 \leq t \leq n-1.$$

where it is common to use $f^n(x_0)$ to denote the composition of $f$ with itself $n$ times, evaluated at point $x_0$. In particular, $C := \{x_0, f(x_0), f(f(x_0)), \ldots, f^{n-1}(x_0)\}$ has $n$ distinct elements (each of which is a point of period $n$) and is called a cycle (or orbit) with period $n$.

**Period 3 implies Chaos**   Of particular interest, is the case of functions $f$ that contain period 3. In 1975, a seminal paper by Li and Yorke (1975), introduced the term "chaos" as used in mathematics nowadays, and proved connections with periodic functions as defined above.

**Fact 1.** *Let $f : \mathbb{R} \to \mathbb{R}$ be continuous. If $f$ has a period 3 point then two properties hold: First, $f$ contains points of any period $n \in \mathbb{N}$. Second, $f$ will exhibit scrambling as defined in Def. 2.1,2.2.*

Sometimes we say that $f$ is "chaotic in the sense of Li and Yorke". On a historical note, this turned out to be a special case of an older result by Sharkovsky (1964).

In other words, a sufficient condition to obtain scrambling phenomena is to show that $f$ contains a point of period 3. In general, the uncountable set of chaotic points may, however, be of measure zero, in which case the map is said to have unobservable chaos or unobservable nonperiodicity (Collet and Eckmann, 2009). Our Theorem 2.4 informs us that this is not the case for randomly initialized RNNs.

**Other RNN Models**   Recurrent models have been thoroughly studied because of their success in sequence learning. Following notation from Laurent and von Brecht (2016), here is a general model for RNNs:

$$u_t = \Phi(u_{t-1}, W_1 x_t, W_2 x_t, \ldots, W_k x_t)$$

where the index $t \in N$ represents the current time, $u_t \in \mathbb{R}^d$ represents the current state of the system, $x_t$ the current external input, and $W_i$'s are weight matrices. Regarding training or initialization of weights many tricks have been proposed, e.g., the popular "identity matrix" trick of Le et al. (2015) to preserve lengths. Laurent and von Brecht (2016) to simplify the above, they study what is referred to as the *dynamical system induced* by the RNN, i.e, they consider trajectories of $u_t$ in the absence of external inputs:

$$u_t = \Phi(u_{t-1}, 0, 0, \ldots, 0)$$

This is much more tractable and gives us insights into the architecture of the RNN as it decouples the influence of the external data $x_t$'s (with which any possible response can be produced) from the model itself. For more on other simplified RNN variants with specific statistical assumptions on weights/input, and connections to edge of chaos, see Bertschinger and Natschläger (2004). Similarly, our $\mathcal{RNN}$ family has no external input; we show that even in 1-dimensional settings, chaos emerges.

## 4   Persistence of Chaos and Phase Transitions

In this section we prove our main results. As we alluded to in the previous section, in order to show chaos in the sense of Li and Yorke, we will bound the probability that the family $\mathcal{RNN}(k, \sigma^2)$ contains a point of period 3, since this will serve as a "witness of chaos". We start with some simple observations.

### 4.1   Triangular Wave with Period 3

Perhaps the simplest example of a continuous function $f$ that has period 3 is the following triangle wave function: $f(x) = 2x$, for $x \in [0, 1/2]$, and $2(1 - x)$, for $x \in (1/2, 1]$. Indeed, one can check that the set $\{\frac{2}{9}, \frac{4}{9}, \frac{8}{9}\}$ is a periodic orbit of length 3, since $f(\frac{2}{9}) = \frac{4}{9}, f(\frac{4}{9}) = \frac{8}{9}$ and finally, $f(\frac{8}{9}) = \frac{2}{9}$ closing the loop. This simple function has played an interesting role in obtaining depth vs width tradeoffs in recent years Telgarsky (2015, 2016); Chatziafratis et al. (2020b).

### 4.2   Persistence of Chaos

Now consider a function from $\mathcal{RNN}$: $f_k(x) = \text{clip}(\sum_{i=1}^k a_i \text{ReLU}(x - b_i))$ for independent $a_i \sim \mathcal{N}(0, \frac{\sigma^2}{k})$ and $b_i \sim \text{Unif}([0, 1])$.

**Theorem 4.1.** *Fix any $\delta \in (0, 1)$. There exists $\sigma$ such that for any sufficiently large $k$ (whose minimum value depends on $\delta$), $f_k$ is 3-periodic with probability at least $1 - \delta$.*

For the theorem we need some intermediate lemmas. We begin with a sufficient condition for chaos:

**Lemma 4.2.** *Let $y_i = \sum_{\iota=1}^{i-1} a_\iota (b_i - b_\iota)$ so that $f_k(b_i) = \text{clip}(y_i)$. If there exists indices $i < \ell$ such that $y_i > 1$ and $y_\ell < 0$, then $f_k$ has period 3.*

*Proof.* We can assume without loss of generality that $b_1 < b_2 < \cdots < b_k$. Suppose there are indices $i < \ell$ as specified in the statement. Since $y_i > 1$, after clipping we know that $f_k(x_i) = 1$, and since $y_\ell < 0$, after clipping we know that $f_k(x_\ell) = 0$. Let's check how intervals are mapped under $f$: since $x_i \leq x_\ell \in [0, 1]$ and because of continuity of $f_k$, we have that $f_k([0, x_i]) = [0, 1]$, and similarly that $f_k([x_i, x_\ell]) = [0, 1]$. By combining this with the intermediate value theorem for $f_k^3$ (3 iterations of $f_k$ applied to itself), we can exhibit a 3-cycle with two elements in $(0, x_i)$ and a third in $(x_i, x_\ell)$. $\qquad\square$

*Proof Sketch of Theorem 4.1.* Roughly, the proof operates by choosing a subset of $t$ ordered elements $\{y_{i_j}\}_{j \in [t]} \subset \{y_i\}_{i \in [k]}$ that meets several key properties such that there exists $j < j'$ such that $y_j > 1$ and $y_{j'} < 0$. Using this and Lemma 4.2 (for $i = j$ and $\ell = j'$), we will conclude that $f_k$ has period 3. However, there are three immediate challenges that make such a result challenging to obtain:

1. (Randomness) If the variance parameter $\sigma$ is small, then there will be a substantial constant probability of an $y_j$ lying in the interval $[0, 1]$ and thus meeting neither characteristic.

2. (Correlations) Due to the function's expression as a linear combination of ReLU neurons with small coefficients, the function $f_k$ is smooth, and $y_i$'s in the same vicinity are highly correlated. This makes it more difficult to ensure that sign changes to the $y_j$'s are likely.

3. (Biases) The randomness of the biases $b_i$ makes it more difficult to assess the correlations of the $y_i$'s, since having small gaps $b_\ell - b_i$ will yield stronger correlations between the two consecutive values of the function.

We mitigate these issues by carefully selecting parameters $t$ and $\sigma$ and the subset $\{y_{i_j}\}_{j \in [t]}$. Problems from (1) can be avoided by choosing $\sigma$ to be large enough to ensure that for sufficiently large $i$ relative to $k$, it's highly unlikely for any $y_{i_j}$ to lie in the interval $[0, 1]$; if none of these lie in that interval, then it is sufficient to find $y_{i_j}$ and $y_{i_{j'}}$ that exhibit a positive-to-negative sign change. To minimize correlation for (2) between nearby $y_{i_j}$'s, we choose those $i_j$'s to ensure that each is a large multiplicative factor times its predecessor; this relies on key intuition from random walks, where increasing temporal differences between random variables makes them increasingly independent. We avoid the atypical samples of $b_i$ of (3) by gradually defining a set of "good" $b_i$'s that exist with high probability and completing the proof while assuming the $b_i$'s are good. $\qquad\square$

*Proof of Theorem 4.1.* A primary goal of our analysis is to bound the variances and covariances of $y_i$'s to aid in our selection of a subset of $y_{i_j}$'s that are nearly independent. We first assume that the biases $b_i$ are held fixed, while the weights $a_i$ are chosen at random; we will later prove bounds that hold for all typical biases. We let $\mathbb{E}_a[\cdot]$ denote such an expectation over random $a$ for fixed $b$. For indices $i < \ell$, we have that $\mathbb{E}_a[y_i] = 0$ and also the following:

$$\mathbb{E}_a\left[y_i^2\right] = \sum_{\iota=1}^{i-1} \mathbb{E}\left[a_\iota^2\right] (b_i - b_\iota)^2 = \frac{\sigma^2}{k} \sum_{\iota=1}^{i-1} (b_i - b_\iota)^2 \text{ and } \mathbb{E}_a[y_i y_\ell] = \frac{\sigma^2}{k} \sum_{\iota=1}^{i-1} (b_i - b_\iota)(b_\ell - b_\iota).$$

For fixed $b_i$'s, each $y_i = \sum_{\iota=1}^{i-1} a_\iota (b_i - b_\iota)$ is drawn from a Gaussian distribution with mean 0 and variance $\frac{\sigma^2}{k} \sum_{\iota-1}^{i-1} (b_i - b_\iota)^2$. We bound these quantities with high probability over random choices of biases $b_i$; for the remainder of the argument, we assume that the low-probability "bad case" (i.e., having small gaps $b_\ell - b_i$) does not occur and apply those bounds without worry about the effects of the bias terms. By defining $b_i$ as the $i$th smallest element from sample of size $k$ from $\text{Unif}([0, 1])$, the distribution of each is a Beta distribution: $b_i \sim \text{Beta}(i, k + 1 - i)$ (see Bertsekas and Tsitsiklis (2008)). We recall the low-order moments of these biases in order to bound the moments of the $y_i$'s. Assume $\iota < i$.

$$\mathbb{E}[b_i] = \frac{i}{k+1}, \quad \mathbb{E}\left[b_i^2\right] = \frac{i(i+1)}{(k+1)(k+2)} \text{ and } \text{Var}[b_i] = \frac{i(k+1-i)}{(k+1)^2(k+2)}$$

We establish a simple lower bound on the summation used to express $\mathbb{E}_a\left[y_i^2\right]$: $\mathbb{E}_a\left[y_i^2\right] \leq \frac{i\sigma^2}{2k}(b_i - b_{i/2})^2$. We apply Chebyshev's inequality to lower-bound $b_i$ and upper-bound $b_{i/2}$. To do so, note

that for sufficiently large $k$, $\mathrm{Var}\,[b_i] \leq \frac{(k+1)^2/4}{(k+1)^2(k+2)} \leq \frac{1}{2k}$. Assume $i \geq k^{3/4}$. Then, by Chebyshev's bound,

$$\Pr\left[b_i \leq \frac{7i}{8k}\right] \leq \Pr\left[|b_i - \mathbb{E}\,[b_i]| \geq \frac{i}{8k}\right] \leq \frac{32}{\sqrt{k}}.$$

By the same calculation, $\Pr\left[b_{i/2} \geq \frac{5i}{8k}\right] \leq \frac{32}{\sqrt{k}}$. Then, $\Pr\left[\mathbb{E}_a\,[y_i^2] \geq \frac{i^3\sigma^2}{8k^3}\right] \geq 1 - \frac{64}{\sqrt{k}}$. Later, this will be important for showing that $y_j \notin [0,1]$ with high probability with $\sigma$ large enough.

We also upper-bound the covariances. Note that $\mathbb{E}_a\,[y_i y_\ell] \leq \frac{i\sigma^2}{k} b_i b_\ell$. For $i \geq k^{3/4}$, the above calculations ensure that $\mathbb{E}_a\,[y_i y_\ell] \leq \frac{3i^2\ell\sigma^2}{2k^3}$ with probability at least $1 - \frac{64}{\sqrt{k}}$.

Now, we choose the elements $y_{i_j}$ that guarantee the needed properties. Let $i_j := \frac{k}{r^{t-j}}$ (assuming that $k$ is divisible by $2r^t$) for $t := \frac{6}{\delta}$ and $r = (12t)^8$. We assume that $k \geq (\frac{72}{\delta})^{192/\delta} = r^{4t}$, which ensures that every $i_j \geq k^{3/4}$, since

$$i_j > \frac{k}{r^t} \geq \frac{k^{3/4}r^t}{r^t} = k^{3/4}.$$

These choices ensure that the variance of each element is high enough to make belonging to $[0,1]$ unlikely for any $y_{i,j}$, while spreading apart successive iterates $y_{i,j}$ and $y_{i,j+1}$ to make them nearly independent. To do so, we analyze the normalized variables $x_j := \frac{y_{i_j}}{\sqrt{\mathrm{Var}_a[y_{i_j}]}}$, for $j \in [t]$.

Note that, given fixed $b$, $x_j \sim \mathcal{N}(0,1)$ and with probability $1 - \frac{128t^2}{\sqrt{k}}$, for every $j < j'$,

$$\mathbb{E}_a\,[x_j x_{j'}] = \frac{\mathbb{E}\left[y_{k/r^{t-j}} y_{k/r^{t-j'}}\right]}{\sqrt{\mathrm{Var}_a\left[y_{k/r^{t-j}}\right]\mathrm{Var}_a\left[y_{k/r^{t-j'}}\right]}} \leq \frac{\frac{3\sigma^2}{2r^{2(t-j)+(t-j')}}}{\frac{\sigma^2}{8r^{3/2(t-j)+3/2(t-j')}}} = 12r^{j/2-j'/2} \leq \frac{12}{\sqrt{r}} \leq \frac{1}{t^4}.$$

Hence, for a sufficiently large $r$, the $x_j$'s are essentially uncorrelated. Note that $x = (x_1, \ldots, x_t) \in \mathbb{R}^t$ (if we assume a fixed $b$ that satisfies the above high probability events) has $x \sim \mathcal{N}(0, \Sigma)$, where $\Sigma_{j,j} = 1$ and $\Sigma_{j,j'} \leq \frac{1}{t^4}$. The smallest eigenvalue of $\Sigma$ can be bounded. (Similarly for $\mu_{\max}(\Sigma)$, see Lemma B.1.) Given the lack of correlation between the $t$ random variables for sufficiently large $r$, we bound the probability that the signs of the elements of $x$ follow a certain pattern. We basically need to show that $x$ has a $+$ sign that precedes a $-$ sign (like triangle waves have $+2$ slope followed by $-2$ slope). Using this, we can estimate the probability of period 3 emerging and this completes the proof. (Due to space limitations, some last steps are in Appendix B.) $\square$

**Corollary 4.3.** *For weights drawn as $a_i \sim \mathcal{N}(0, \omega(\frac{1}{k}))$, the $\lim_{k \to \infty} \Pr\,[f_k \text{ is 3-periodic}] = 1$.*

Combining the fact that $f_k$ will have period 3 with some constant probability $\delta$, with the work of Chatziafratis et al. (2020b) on the expressivity of networks as measured by #linear regions we get:

**Corollary 4.4.** *With the same probability $\delta$, the number of linear regions of $f_k \in \mathcal{RNN}$ is exponential in the number $t$ of feedback iterations, with a growth rate of $\phi = \frac{1+\sqrt{5}}{2}$.*

**Theorem 4.5.** *Consider $f_k$ initialized according to the He initialization (i.e. $f_k$ from Thm. 4.1 with $\sigma = 2$). For sufficiently large $k$, $f_k$ is 3-periodic w.p. at least some constant. (Proof in App. B.2.)*

**Theorem 4.6.** *Consider $f_k$ initialized with $a_i \sim \mathcal{N}(0, \frac{1}{4k\log k})$. Then, the probability that $f_k$ has no fixed points (therefore exhibits no chaotic behavior) is $\geq 1 - \frac{1}{k}$. (Proof in App. B.3.)*

## 5 Experiments

Our goal is to evaluate how robust our findings are to specifications of the RNN model, and to empirically estimate some bounds on how often scrambling phenomena appear[2].

---

[2]Our code is made publicly available here: `https://github.com/steliostavroulakis/Chaos_RNNs/blob/main/Depth_2_RNNs_and_Chaos_Period_3_Probability_RNNs.ipynb`

**Robustness across Different Models**  We try different architectures and initialization techniques from the literature ([He et al.](#), 2015; [Glorot and Bengio](#), 2010), truncated Gaussians, and others, for setting the weights and the biases. Our table in Fig. 3 summarizes the results across 10000 runs of each experiment. For each experiment, the first layer has width 2, where the weights and biases of each neuron are initialized according to each line, whereas the second layer has width 1 with weight and bias as specified in each line. Notice that the last column contains the probability that the recurrent model is chaotic, having period 3. The way we classify whether the output of an RNN is chaotic or not is done by plotting one iteration of the RNN and three iterations of the RNN and then checking whether there are more fixed points in the latter, implying period 3 (for a pictorial representation see Figure 7 in Appendix C). The purpose of this experiment was to quantify the probability for

| Layer 1 | | | Layer 2 | | | Pr[period 3] |
|---|---|---|---|---|---|---|
| weight $w$ | bias $b$ | activation | weight $w$ | bias $b$ | activation | |
| 1 | He | ReLU | He | He | 1 | **13.77%** |
| He | Uniform | ReLU | He | Uniform | ReLU | **7.49%** |
| He | He | ReLU | He | He | ReLU | **4.51%** |
| $\mathcal{N}\left(0,\frac{1}{k}\right)$ | $\mathcal{N}\left(0,\frac{1}{k}\right);[-1,1]$ | ReLU | $\mathcal{N}\left(0,\frac{1}{k}\right)$ | $\mathcal{N}\left(0,\frac{1}{k}\right);[-1,1]$ | ReLU | **4.45%** |
| Glorot | Glorot | ReLU | Glorot | Glorot | ReLU | **2.28%** |

Figure 3: The rightmost column has the estimates for the probability that the RNN exhibits period 3. We ran the experiment for 10000 times and checked whether the random RNN has period 3 (see Fig. 7). Each line specifies the type of initialization or activation unit used.

having period 3 in one-dimensional randomly initialized RNNs, as in higher dimensions there is not a clean mathematical statement for checking periods 3. This validates our theoretical findings that chaos (in the sense of Li-Yorke) persists, in contrast to FNNs (see Figure 2).

**RNNs transitions in Higher Dimensions**  In higher dimensions, we used an MNIST dataset as input to a 64-dimensional RNN with 1 hidden layer, width 64, fully connected, with ReLU activation functions and He initialization. Note that we did not train the RNN as this was not our purpose. We simply initialized it and observed (Fig. 4) how the values of the output neurons fluctuate with respect to $t$ (the number of compositions of the RNN with itself) for various $\sigma$. Since high dimensions do not have clean characterizations of chaos such as period 3, we instead detect chaos by relying on a necessary condition: Jacobian's spectral norm being more than 1 (See Fig. 5).

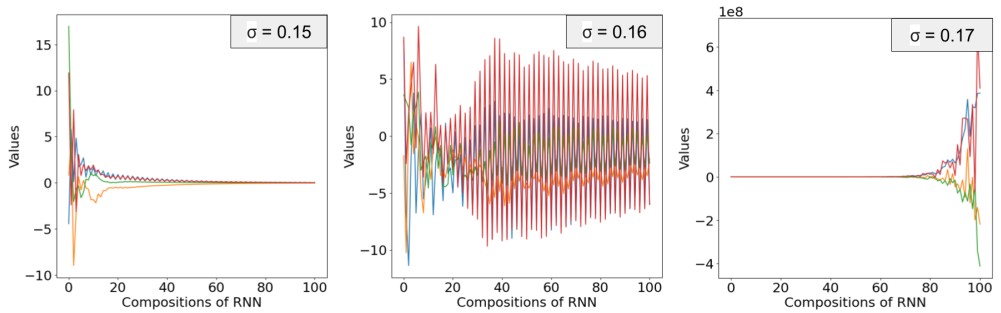

Figure 4: The y-axis has values for 4 different output neurons, while x-axis is number of RNN compositions. From left-to-right, as we vary $\sigma$, RNN becomes unstable similarly to 1D case in Fig. 1.

**Additional Experiments and Remarks.**  Regarding the experiments reported in Figure 3, the reported numbers were without the Clipping operation, so that the architecture is more faithful to practice. We believe it is an interesting direction to further understand the chaotic properties of neural networks at initialization and exactly quantify both experimentally and theoretically the probability of chaos. Of course, for the simpler one-dimensional case, we were able to use the period 3 property which is a simple to state and simple to verify condition. For higher dimensional RNNs, that are closer to those being used in practice nowadays, it still remains a challenging problem to characterize when chaos emerges. Other notions of chaos like Devanay's chaos ([Huang and Ye](#), 2002) could also

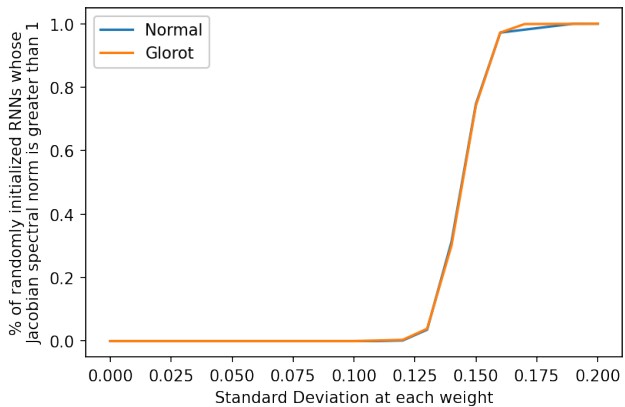

Figure 5: The y-axis is the % of RNNs (out of 1000 runs) having large Jacobian norm, and x-axis is varying values of $\sigma$. Echoing our main theorems 2.4, 2.5, we experimentally determine for two popular initialization schemes the threshold where the input-output Jacobian has spectral norm $> 1$.

be useful when exploring chaotic properties of neural nets (Devaney's chaos is a stronger notion of chaos compared to Li–Yorke's chaos).

For completeness, we also performed some additional experiments with the Clipping operation and the numbers are still ranging from 2%-9% for the probability of chaos. Specifically, following the setup of Fig. 3, out of 10000 runs, we got (with Clipping) that the chaos frequency reported in the last column is $3.6, 5.8, 9.7, 5.7, 2.2$ respectively for the different initializations. We also performed experiments with other types of activations units and higher values for the width and the depth and we still observe that these variants too exhibit period 3. For example, with tanh activations, the last column would be $9.7, 11, 17.6, 2.6, 4.6$ respectively. If we instead change the width to be equal to 4, the numbers become $13.1, 8.1, 14.09, 0.8, 2$ and in another setting where the depth equals 3 the numbers become $6.5, 3.1, 5, 0.34, 1.14$.

## Conclusion

Using tools from discrete dynamical systems, we show simple models for randomly initialized recurrent neural networks exhibiting scrambling phenomena and we study the associated threshold phenomena for the emergence of Li-Yorke chaos. Our theory and experiments explain observed RNN behaviors, highlighting the difference between FNNs and RNNs in a simple setting. More broadly, we believe this connection can be fruitful to obtain a better understanding of RNNs at initialization as it adds to a recent thread of works that give interesting new perspectives on neural networks, borrowing ideas from dynamical systems.

## Acknowledgments

The authors would like to thank the anonymous reviewers for their time and their useful feedback that improved the presentation of our paper. VC would like to acknowledge support from a start-up grant at UC Santa Cruz. IP would like to acknowledge support from a start-up grant at UC Irvine. Part of this work was done while the authors VC, IP, SS were visiting UC Berkeley and the Simons Institute for the Theory of Computing, while VC was a FODSI fellow at MIT and Northeastern. CS was supported by NSF GRFP and NSF grants CCF-1814873 and IIS-1838154.

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
