## A An additional example

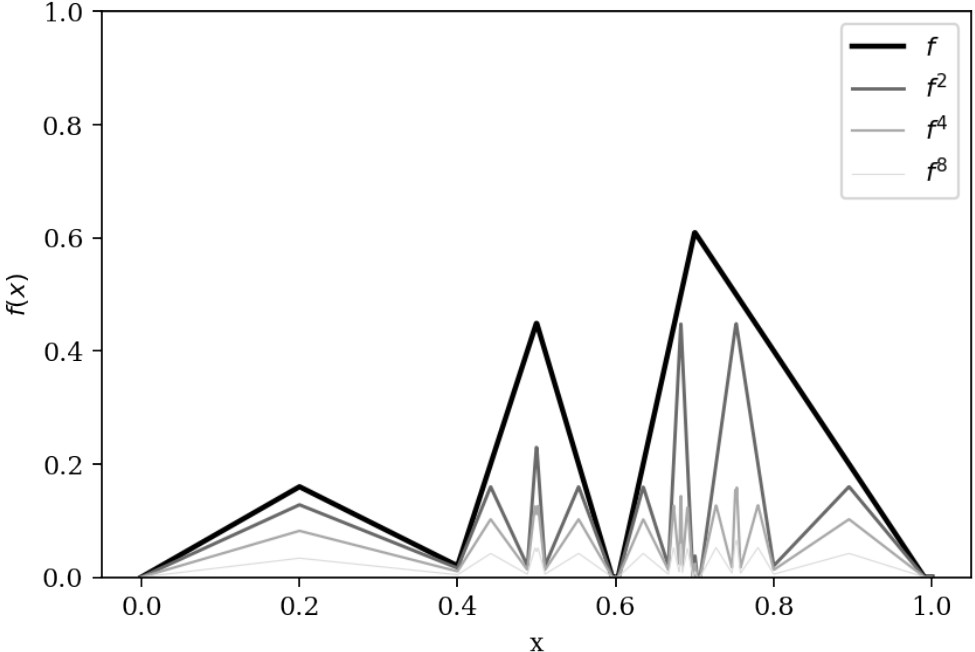

Figure 6: A function $f$ and its repeated compositions $f^2(x), f^4(x), f^8(x)$. Despite the large Lipschitz constant $L > 6$ (notice here this coincides with the spectral norm of the Jacobian as well, since $f$ is piecewise linear function), there will be no scrambling phenomena. Indeed, $f$ is a contraction or a contractive map, and repeated applications of $f$ will lead to the zero function, so no scrambled sets exist. Moreover, since the function does not have any periodic points (other than fixed points), the number of linear regions cannot grow exponentially (see also Chatziafratis et al. (2020b) where they prove that *odd* periods (larger than 1) are those that cause exponentially many linear regions).

The definitions of scrambling and scrambled sets (2.1) capture exactly the fact that trajectories of nearby points can get arbitrarily close, yet also move apart infinitely often (as in Fig. 1). Intuitively, when scrambling occurs in RNNs, their number of linear regions created by the output neurons will grow large with the number of iterations and their input-output Jacobian will have spectral norm larger than one. Concretely, see the work of Chatziafratis et al. (2020b), where they quantify exactly the rate of growth for the linear regions based on periodic points of the function; for example, if the function has period 3 then the number of linear regions grows exponentially in the number of iterations of the RNN (the base of exponent is the golden ratio $\approx 1.618$) and the Jacobian of the network has spectral norm at least as large as 1.618.

However, the opposite is not true, as there could be examples (see Figure 6), where the spectral norm of the Jacobian is large, even though there is neither scrambling phenomena nor high expressivity.

## B Omitted proofs

### B.1 Proof of Theorem 4.1 and supporting lemmas

**Lemma B.1.** *The smallest (and largest) eigenvalue of $\Sigma$ can be bounded. Specifically, $\mu_{\min}(\Sigma) \geq 1 - \frac{1}{t^3}$ and $\mu_{\max}(\Sigma) \leq 1 + \frac{1}{t^3}$.*

*Proof.*

$$\mu_{\min}(\Sigma) \geq \min_x \frac{x^T \Sigma x}{\|x\|_2^2} \geq \min_x \frac{1}{\|x\|_2^2} \sum_{j=1}^t \left( x_j^2 - \sum_{j' \neq j} \frac{1}{2t^4}(x_j^2 + x_{j'}^2) \right)$$

$$\geq \min_x \frac{1}{\|x\|_2^2} \left( \|x\|_2^2 - \frac{1}{t^3}\|x\|_2^2 \right) = 1 - \frac{1}{t^3}.$$

Similarly, we have $\mu_{\max}(\Sigma) \leq 1 + \frac{1}{t^3}$. $\qquad\square$

Given the lack of correlation between the $t$ random variables for sufficiently large $r$, we bound the probability that the signs of the elements of $x$ follow a certain pattern. That is, fix any $\alpha \in \{-1, 1\}^t$. Our goal is to lower-bound the probability that $\text{sign}(x_j) = \alpha_j$ for all $j$ to be just smaller than $\frac{1}{2^t}$. Note that $\det(\Sigma) \geq \mu_{\min}(\Sigma)^t \geq (1 - \frac{1}{t^3})^t \geq 1 - \frac{1}{t^2}$. Similarly, $\det(\Sigma) \leq \mu_{\max}(\Sigma)^t \leq 1 + \frac{1}{t^2}$.

**Lemma B.2.** *Let $\mathbb{R}^t_\alpha$ be the orthant having $\text{sign}(x_j) = \alpha_j$ for all $x \in \mathbb{R}^t_\alpha$. Then*

$$\Pr\left[ \forall j \in [t], \ \text{sign}(x_j) = \alpha_j \right] \geq \left( 1 - \frac{1}{t} \right) \cdot \frac{1}{2^t}$$

*Proof.*

$$\Pr\left[ \forall j \in [t], \ \text{sign}(x_j) = \alpha_j \right] = \int_{\mathbb{R}^t_\alpha} \frac{1}{(2\pi)^{t/2}\sqrt{\det(\Sigma)}} \exp\left( -\frac{1}{2} x^T \Sigma^{-1} x \right) dx$$

$$\geq \int_{\mathbb{R}^t_\alpha} \frac{1}{(2\pi)^{t/2}\sqrt{1 + \frac{1}{t^2}}} \exp\left( -\frac{1}{2} \|x\|^2 \mu_{\max}(\Sigma^{-1}) \right) dx$$

$$\geq \sqrt{1 - \frac{1}{t^2}} \int_{\mathbb{R}^t_\alpha} \frac{1}{(2\pi)^{t/2}} \exp\left( -\frac{\|x\|^2}{2\mu_{\min}(\Sigma)} \right) dx$$

$$\geq \sqrt{1 - \frac{1}{t^2}} \left( 1 - \frac{1}{t^2} \right)^{t/2} \int_{\mathbb{R}^t_\alpha} \frac{1}{(2\pi)^{t/2}\left(1 - \frac{1}{t^2}\right)^{t/2}} \exp\left( -\frac{\|x\|^2}{2(1 - \frac{1}{t^2})} \right) dx$$

$$= \left( 1 - \frac{1}{t^2} \right)^{(t+1)/2} \Pr_{x \sim \mathcal{N}(0, \frac{1}{1-1/t^2} I_t)}\left[ \forall j \in [t], \text{sign}(x_j) = \alpha_j \right]$$

$$\geq \left( 1 - \frac{1}{t} \right) \cdot \frac{1}{2^t}.$$

$\qquad\square$

Instead of finding the probability that $x$ has a certain *fixed* sequence of signs, according to Lemma 4.2, we need to show that it has a positive sign that precedes a negative sign.

**Definition B.3.** We say that $x$ is "good" if the sequence of signs corresponding to $x$ has a positive sign that precedes a negative sign. (This corresponds to the requirement that there is some $i < \ell$, such that $y_i > 1$ and $y_\ell < 0$, although we haven't yet mandated that $y_i > 1$; only $y_i > 0$.)

Note that this is a property of all but $t + 1$ of the $2^t$ combinations of signs $\alpha$. (The only $\alpha$'s without a negative sign following a positive one are those consisting of all negative $\alpha_j$'s followed by positive $\alpha_j$'s, whose number is only $t + 1$.

$$\Pr_a\left[ x \text{ is good} \right] \geq (2^t - t - 1)\left( 1 - \frac{1}{t} \right) \cdot \frac{1}{2^t}$$

$$= \left( 1 - \frac{t}{2^t} - \frac{1}{2^t} \right)\left( 1 - \frac{1}{t} \right) \geq 1 - \frac{\delta}{3}.$$

As a result, we can conclude that $f_k$ is 3-periodic so long as $x$ is good, the $b_j$'s are well-behaved (with probability at least $1 - \frac{96}{\sqrt{k}} \geq 1 - \frac{\delta}{3t}$), $y_{k/r^{t-j}} \notin [0, 1]$ for all $j \in [t]$) and $y_{k/r^{t-j}} \geq 1$ if $\alpha_j = 1$ and $y_{k/r^{t-j}} \leq 0$ if $\alpha_j = -1$. To bound that probability, we place an upper-bound on the probability that any $y_{k/y^{j-1}} \in [0, 1]$ and assume that $\sigma \geq \frac{tr^{3t}}{\delta}$.

**Lemma B.4** (Conclusion of proof of Theorem 4.1). $\Pr\left[f_k \text{ is 3-periodic}\right] \geq 1 - \delta$

*Proof.*

$$
\begin{aligned}
\Pr\left[f_k \text{ is 3-periodic}\right] &\geq 1 - \frac{\delta}{3} - \frac{96t}{\sqrt{k}} - t\Pr_a\left[y_{r^{t-1}} \in [0,1]\right] \\
&\geq 1 - \frac{\delta}{3} - \frac{\delta}{3} - \frac{t}{\sqrt{2\pi\mathbb{E}_a\left[y_{k/r^{t-1}}^2\right]}} \\
&\geq 1 - \frac{2\delta}{3} - \frac{t}{\sqrt{2\pi \cdot \frac{3(k/r^{t-1})^3\sigma^2}{2k^3}}} \\
&\geq 1 - \frac{2\delta}{3} - \frac{tr^{3t}}{3\sigma} = 1 - \delta.
\end{aligned}
$$

$\square$

## B.2 Proof of Theorem 4.5

**Theorem 4.5.** *Consider $f_k$ initialized according to the He initialization (i.e. $f_k$ from Thm. 4.1 with $\sigma = 2$). For sufficiently large $k$, $f_k$ is 3-periodic w.p. at least some constant. (Proof in App. B.2.)*

*Proof.* The proof echoes that of Theorem 4.1. We consider the formulation of the previous problem with $t = 2$ and $r = 288$. That is, $y_{i_1} = y_{k/288}$ and $y_{i_2} = y_k$.

By the previous proof, with probability at least $1 - \frac{128}{\sqrt{k}}$, $\mathbb{E}_a\left[y_{k/288}^2\right] \geq \frac{1}{2 \cdot 288^2}$ and $\mathbb{E}_a\left[x_1 x_2\right] \leq \frac{1}{\sqrt{2}}$ for $x_j := \frac{y_{i_j}}{\sqrt{\mathrm{Var}_a[y_j]}}$. We say that this event occurs if "$b$ is good." We can then place a lower bound on the probability that $f_k$ is 3-periodic. Let $\Sigma \in \mathbb{R}^{2\times 2}$ be a covariance matrix with $\Sigma_{1,1} = \Sigma_{2,2} = 1$ and $\Sigma_{1,2} = \Sigma_{2,1} = \frac{1}{\sqrt{2}}$.

$$
\begin{aligned}
\Pr\left[f_k \text{ 3-periodic}\right] &\geq \Pr\left[b \text{ is good}, y_{i_1} \geq 1, y_{i_2} \leq 0\right] \\
&\geq \Pr\left[y_{i_1} \geq 1, y_{i_2} \leq 0\right] - \frac{128}{\sqrt{k}} \\
&\geq \Pr\left[x_1 \geq 2 \cdot 288^2, x_2 \leq 0\right] - \frac{128}{\sqrt{k}} \\
&\geq \Pr_{g \sim \mathcal{N}(0,\Sigma)}\left[g_1 \geq 288^2, g_2 \leq 0\right] - \frac{128}{\sqrt{k}} \\
&:= c - \frac{128}{\sqrt{k}} \geq \frac{c}{2}.
\end{aligned}
$$

The second-last inequality relies on the fact that $\Pr_{g \sim \mathcal{N}(0,\Sigma)}\left[g_1 \geq 288^2, g_2 \leq 0\right]$ is an extremely small constant $c$, and the final one assumes that $k \geq \frac{2^{15}}{c^2}$.

$\square$

## B.3 Proof of Theorem 4.6

**Theorem 4.6.** *Consider $f_k$ initialized with $a_i \sim \mathcal{N}(0, \frac{1}{4k \log k})$. Then, the probability that $f_k$ has no fixed points (therefore exhibits no chaotic behavior) is $\geq 1 - \frac{1}{k}$. (Proof in App. B.3.)*

*Proof.* To show that $f_k$ has no fixed points, it suffices to show that $y_i < b_i$ for all $i \in [k]$. This ensures that $f_k(x) \leq x$ for all $x \in (0, 1]$, which means that $f_k^t(x)$ approaches zero as $t$ increases and ensures that there are no fixed points.

From the proof of Theorem 4.1, $y_i \mid b_i$ is a Gaussian random variable with mean 0 and variance

$$\frac{1}{4k \log k} \sum_{\iota=1}^{i-1} (b_i - b_\iota)^2 \frac{i b_i^2}{4k \log k} \le \frac{b_i^2}{4 \log k}$$

for any $0 < b_1 < \cdots < b_k < 1$. We now show that the probability that any $y_i \ge b_i$ is small by using Gaussian tail bounds.

$$\Pr\left[f_k \text{ has a fixed point}\right] \le \Pr\left[\exists i \in [k] \text{ s.t. } y_i \ge b_i\right] \le \sum_{i=1}^{k} \Pr\left[y_i \ge b_i\right] \le \sum_{i=1}^{k} \sup_b \Pr\left[y_i \ge b_i \mid b_i\right]$$

$$\le \sum_{i=1}^{k} \sup_b \exp\left(\frac{b_i^2}{2\mathrm{Var}\left[y_i^2 \mid b_i\right]}\right) \le \sum_{i=1}^{k} \exp\left(2 \log k\right) \le \frac{1}{k}.$$

$\square$

# C Omitted figure in experiments

In the experimental section, Sec. 5, we focused on randomly initialized RNNs and we need to check their periodicity, specifically whether or not the output function is chaotic or not (in particular whether or not it had period 3, and hence is Li-Yorke chaotic). But how can we do this?

The way we classify whether the output of an RNN is chaotic or not is done by plotting one iteration of the RNN and three iterations of the RNN and then checking whether there are more fixed points in the latter, implying period 3. See Fig. 7b, where the blue plot has many more intersections with the identity line $y = x$ compared to Fig. 7a. Such intersection points that do not constitute fixed points of $f(x)$ are periodic points of period 3.

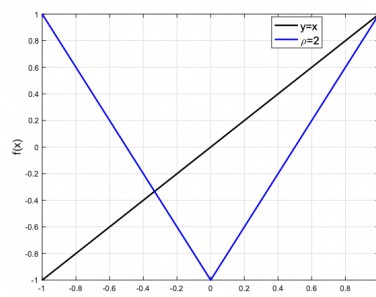

(a) Graph of $f(x)$ intersected with $y = x$, to identify period 1 points.

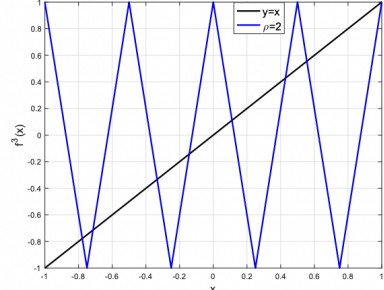

(b) Graph of $f^3(x)$ intersected with $y = x$, to identify period 3 points.

Figure 7: We plot the output of RNN, after 1 and 3 feedback loops. Intersections with $y = x$ inform us about periodicity. The plot on the right is $f^3$ and the fact that there are more fixed points of $f^3$ than for $f$ (plot on the left) means that there is at least one periodic point with period 3, which means $f$ exhibits Li-Yorke chaos.