# OpenReview forum: "On Scrambling Phenomena for Randomly Initialized Recurrent Networks "
_NeurIPS.cc/2022/Conference — NeurIPS 2022 Accept_

### Official Review · Reviewer_YhQb · 2022-07-04

**Rating:** 5
**Confidence:** 2
**Soundness:** 3 good
**Presentation:** 2 fair
**Contribution:** 3 good

**Summary:**

The paper explores how randomly initialized RNNs exhibit "Li-Yorke chaos" with a certain probability and which leads to exploding gradients and other undesirable phenomena. By using the fact that functions containing a period three point it exhibits chaos, some probabilistic bounds on the emergence of chaos in RNNs are derived. Finally, the theoretical results are evaluated with two toy examples.

**Questions:**

- Is the resulting chaos actually desirable or should it be avoided when possible?
- If it should be avoided, what alternative initialization procedure could be used?
- What would be the theoretical bounds for the Probabilities given in Figure 3?

**Limitations:**

Not applicable IMO

**Strengths And Weaknesses:**

Strengths:
- Stunning maths
- Interesting little experiments with code attached

Weaknesses:
- Short conclusion, no immediately apparent benefit other than a new name for the phenomenon.

Suspected Errors:
- Figure 1: A legend would be helpful -  what is the blue line? Typo in caption: "adabehind"
- Figure 2: Unclear how spikes in the plots correspond to linear regions
- Lemma 4.2.: Some y's should be x's?
- Line 156 end: something missing here

---

> ### Author Response · Authors · 2022-08-02
> **Addressing Questions**
>
> We thank the reviewer for the feedback and interesting questions. We will fix all minor issues as suggested. Below we address their main questions.
>
> --Desirability of Chaos:  To the question about the desirability of chaos, the answer is that there is a well-documented tradeoff between expressivity and chaos. In particular, lines 168-176 they point to related works (e.g., [1] and more) where the ideal regime is the so-called ''edge of chaos''. To make it clear, of course we can avoid chaos by initializing everything to zero but then vanishing gradient problem would prevent learnability. On the other hand, having chaotic behavior can lead to exploding gradients which is also a problem. The ''edge-of-chaos'' studies regimes where the network is somewhat stable by having (in some sense) an input-output Jacobian whose norm is close to 1.  Moreover, the tradeoff becomes apparent with respect to expressivity too. RNNs in the chaotic regime are much more likely to suffer from the exploding gradient problem and be hence much more difficult to train. However, there are clear tradeoffs as evidenced by past papers, which illustrate the more chaotic models (as evidenced by higher Jacobians or the existence of Li-Yorke chaos) have greater expressive properties [1,2]. Intuitively, being too far away from the chaotic regime causes a greater likelihood of the vanishing gradient issue and of inexpressive initializations. We believe understanding such phenomena is of great interest to the community and that our paper builds towards this direction.
>
> [1] Mean Field Residual Networks: On the Edge of Chaos by Greg Yang, Samuel S. Schoenholz\
> [2] Exponential expressivity in deep neural networks through transient chaos by Poole, B., Lahiri, S., Raghu, M., Sohl-Dickstein, J., & Ganguli, S. (NeurIPS 2016)
>
> --On Exact Probability Values: Finding exact numbers for the probability of having period 3 in RNNs is not an easy task. We can compare the experimental values for the probability of chaos as reported in Figure 3 with our theoretical values though. It is easy to see that our analysis predicts lower bounds for the probability of chaos that are smaller than the empirical values. In the analysis of our random walks, it is difficult to keep track of all the constants and so our lower bounds for the emergence of chaos are slightly pessimistic. This was part of the reason we wanted to perform the experiments to observe how often will chaos emerge. By the last column of Figure 3, we see that standard methods for random initilization of RNNs such as He initialization, Glorot initialization etc. give rise to chaotic behavior with some good prrobability ~4-5%. Part of our contribution is the fact that we proved such chaotic behavior for standard methods of initialization.
>
> --Applicability: Regarding the question about the applicability of the increased understanding of chaotic behavior in RNNs that our paper provides, we believe that our results formalize the difficulty of training RNNs with stability vs FNNs, by demonstrating a marked contrast with the past work by Hanin and Rolnick [1,2]. Taken together, our work and these two papers illustrate that a properly parameterized initialization for RNNs is more critical to get right than for FNNs, since minor changes in parameterization and network width cause RNNs to switch between stable and chaotic regimes. We go beyond past analyses of RNN stability that depend only on the Jacobian by drawing a direct link between the parameter initialization and the scrambling phenomenon. We also use different techniques from [1,2] (as they relied on independent noise for each parameter) and different techniques from prior works relying on mean-field analysis [3] that only hold in the infinite width regime.
>
> [1]: Complexity of Linear Regions in Deep Networks by Boris Hanin, David Rolnick (ICML 2019)\
> [2]: Deep ReLU Networks Have Surprisingly Few Activation Patterns by Boris Hanin, David Rolnick (NeurIPS 2019)\
> [3]: Real-time computation at the edge of chaos in recurrent neural networks by Bertschinger, N. and Natschlager, T. (Neural computation 2004)

---

> > ### Comment · Reviewer_YhQb · 2022-08-08
> > **Thanks for the Answer**
> >
> > Thank you for the rigorous answer. Today I learned something.
> > Best of Luck!

---

### Official Review · Reviewer_niom · 2022-07-11

**Rating:** 6
**Confidence:** 2
**Soundness:** 3 good
**Presentation:** 3 good
**Contribution:** 3 good

**Summary:**

This paper studies the behavior of a specific class of RNNs at initialization using dynamical systems theory. Its main result shows that chaos can arise in RNNs even in the absence of external input and this behavior holds independently of the width. This result explains the scrambling phenomenon observed experimentally. It also shed light on the expressive power of RNNs, in terms of the number of linear regions formed by the nonlinear activations.

**Questions:**

- Is it possible to clarify the connection between Li-Yorke chaos and exploding gradients slightly more precisely?
- Will similar chaotic behavior be observed for other variants of RNNs (say with different activations and depth)?
- For the experiments with MNIST, how does the Pr[period 3] depends on the width of RNN? Would be good to demonstrate some results along this line

**Limitations:**

In the checklist, it was mentioned that limitations are described, but I could not find any such discussions in the main paper.

**Strengths And Weaknesses:**

Strengths
- Provide first understanding of RNNs via the lens of Li-Yorke chaos
- Overall well written

Weaknesses
- Analysis is limited to scalar inputs and RNNs with ReLU activation (tanh activation is often used in practice)
- There are no results connecting the behavior of input data to the chaotic behavior of (input-driven) RNNs, which are highly relevant in practice

---

> ### Author Response · Authors · 2022-08-02
> **Clarifications for Variants of our Model**
>
> We thank the reviewer for their feedback and questions.
>
> --TANH: The reviewer asks what happens if instead of ReLU we had a tanh activation which is commonly used for RNNs. We perform experiments and we see that even with TANH, the RNN will exhibit Li-Yorke chaos with some good probability.
>
> To check this, we performed a set of experiments similar to those shown in Figure 3 of our paper, but this time we used the TANH activation, instead of a ReLU. With tanh activations, we similarly estimate the frequency of period 3 (last column of Fig. 3). Specifically, if instead of ReLU we used tanh activations the last column would be:
>
> (width =2, TANH) Chaos Frequency %\
> 11\
> 9.7\
> 7.6\
> 2.6\
> 4.6\
>
> --Higher width and depth: We also performed experiments where we increased the network's width and depth. Specifically, we used a similar experimental setup as the one shown in Figure 3, where the width was increased to 3 and to 4 and the depth to 3. The frequency of period 3 was slightly changing. For example, for depth 3 the resulting numbers for the last column of Fig. 3 are:
>
> (DEPTH = 3, ReLU) Frequency (out of 10000 runs) of Chaos %\
> 6.5\
> 3.1\
> 5\
> 0.34\
> 1.14
>
> For other values of width, the resulting numbers for the last column of Fig. 3 are:
>
> (WIDTH = 4, ReLU) Frequency (out of 10000 runs) of Chaos %\
> 13.1\
> 8.1\
> 4.09\
> 0.8\
> 2
>
> As we can see, it is not always obvious how the width or the depth affects the probability for chaotic behavior. For example, it is not always monotone as we can see, and presumably there is a dependence on the exact architectural choices made for the activations or the random initialization choices. We believe that rigorously establishing the connection among initialization choices, depth, width and Li-Yorke chaos is an interesting yet challenging question.
>
> --Li-Yorke chaos vs exploding gradients: The implication goes only one-way, i.e., that Li-Yorke chaos implies exploding gradients, but not other way around. For example, in the Appendix A of the submitted version, in Figure 6, we give a relevant example that could help clarify the connection.
>
> --Input-driven RNNs: We refer the reviewer to our discussion in lines 194-201 where we explain the benefits of doing an analysis for the homogeneous RNN model without inputs. For further context, we also refer the reviewer to [1]. Briefly, the RNN system with no input is much more tractable, and it offers a means to investigate the inner working of a given recurrent model because it separates the influence of input data. With input data, the model essentially can produce any possible trajectory on its output. In order to decouple this from the model itself, and study what kinds of trajectories are attainable from the model itself, we have to study RNNs in the absence of input. Our paper studies the random initialization properties for common heuristics of initialization (He, Glorot etc.) of RNNs, and we believe incorporating more information about the input to the system is an interesting direction for future work.
>
> [1]: A recurrent neural network without chaos by Thomas Laurent, James von Brecht (ICLR 2017)

---

### Official Review · Reviewer_EXTU · 2022-07-12

**Rating:** 9
**Confidence:** 4
**Soundness:** 4 excellent
**Presentation:** 4 excellent
**Contribution:** 4 excellent

**Summary:**

The authors establish non-asymptotic probability bounds that a random (randomly initialized) recurrent neural network is chaotic.
The analysis is performed as follows

1. Hidden state $x_t$ is scalar, and the state space is compact interval $x_t \in [a,b] \subset \mathbb{R}$

2. ReLU is used as the activation function, though the analysis only requires continuity rather than differentiability

3. Using the specific definition of chaos due to Li and Yorke for interval maps, the authors need to bound the probability that the RNN $f$ has period three fixed points, i.e. $f^3(x^*) = x^*$

4. The probability of such an event is then expressed in terms of the number of neurons $k$, the variance parameter $\sigma^2$ of a Gaussian distribution from which the 'weight matrix' is sampled.

**Questions:**

> small perturbations do not break expressivity
I am not overwhelmingly familiar with symbolic dynamics, but if the activation were smooth, then the persistence should be guaranteed due to hyperbolicity. I wonder if the authors could offer some comments on the hyperbolicity and robustness.

Lines 190-193:
Could the authors comment further on cases when chaos is unobservable?

Figure 5 and accompanying code:
It appears the figure is produced by calculating the input-output Jacobian of the 2 layer NN specified in the paragraphs 'RNNs transitions in Higher Dimensions'. I would argue that evaluating either input-output Jacobian of the iterated map or its Lyapunov exponent would provide a more convincing evidence for chaos.
On a related note, I do believe that use of the function `complexe_modulo` is erroneous-- stability of the system should depend only on the real part of the eigenvalues.

**Strengths And Weaknesses:**

The submission offers a novel, rigorous analysis that quantifies how commonly is chaos found in RNNs with finitely many neurons.
Unlike the mean-field approaches, $\lim{k \to \infty}$ isn't necessary.

This allows to establish chaos more rigorously than the use of Lyapunov exponents usually allows for.

Overall, I believe this to be an extremely interesting study, with excellent presentation.

I would like to offer some comments, presented in chronological order, that might help improve the manuscript further:

1. I found Figure 2 somewhat hard to understand, at least initially. It might be worthwhile displaying the equation for the tent map in the figure. The graph added little intuition, partly because nodes and edges aren't explained. It might also help to mention that the FNN graph is simply 'unrolled'. Perhaps it would also help to indicate where the small Gaussian perturbation is supplied? What overall effect on the dynamics do stochastic perturbations to $b_i$'s have?

2. Li-Yorke chaos is introduced, but never contrasted with other definitions e.g. one due to Devaney. This could also give an avenue to explicit all the consequences of LY chaos for interval maps, such as sensitive dependence on initial conditions/exploding gradients.

Section 2.1
Most of the cited references analyze their respective problems using ergodic theory. The juxtaposition with LY chaos is entirely warranted, but the phrasing of the sentence gives the impression that non-reliance on LY is somehow 'bad'.

Paragraph 'RNNs transitions in Higher Dimensions':
Invoking LY chaos is warranted for interval maps. For higher dimensional systems period-3 fixed points do not provide sufficient nor necessary conditions for chaos. Similarly, requiring the existence of a scrambled set as in Definition 2.1, can produce dynamics that do not have the sensitive dependence on initial conditions.
I believe this should be acknowledged. In a similar vein, paragraph 'Techniques' (lines 143-159) gives an impression that the analysis is more general than the mean-field one, rather than that it lies at the "opposite end of the spectrum". I believe it should be stated more clearly in the sections of the submission that technical arguments apply only to RNNs with a scalar hidden state $x_t$.

---

> ### Author Response · Authors · 2022-08-02
> **Addressing comments/questions**
>
> We thank the reviewer for the positive feedback and overall support of our paper. We address their comments below and we will include all suggested comments in the final version.
>
> --- For Figure 2: We will clarify further the caption of Fig. 2 and will include the equation for the triangle: (f(x)=2x for x in [0,½], and f(x)=2-2x for x in [½,1]). To add a bit more: the left part of the figure (FNN) is simply a recreation of a figure from Hanin and Rolnick [1, see their Fig. 3] and was presented here for completion. The small Gaussian perturbation is added on every weight and on every bias for each of the layers of the FNN. To put it simply, every number shown in Fig. 2 is perturbed with Gaussian noise.  In contrast, the right part of the figure (RNN) is essentially the same network with the important difference that we add a Gaussian perturbation only to the numbers shown on this right part. Because the network is recurrent, the same weight and bias parameters are used across different iterations, and this turns out to have a significant effect on the number of linear regions, as depicted. Regarding the reviewer's question, we can easily plot the same graphs, in the case that the stochastic perturbation is only added to the biases and not the weights, to see the effect on expressivity.
>
> [1]: Complexity of Linear Regions in Deep Networks by Boris Hanin, David Rolnick (ICML 2019)
>
> --- Devanay Chaos: We thank the reviewer for pointing out to this very interesting direction. We will add a comparison to this type of chaos as suggested. An important result here is the fact that Devaney’s chaos is a stronger notion of chaos compared to Li–Yorke’s chaos [1]. Briefly, we believe the reason behind this is the fact that Devanay chaos requires a ``transitivity'' property for the function under investigation; however for Li-Yorke this is not necessary: for example consider the modified triangle wave f(x)= 0 for x in [0,2/9-$\epsilon$], f(x)=2x for x in [2/9, 1/2], f(x)= 2(1-x) for x in [1/2,8/9] and f(x)=0 for x in [8/9+$\epsilon$,1]. (the $\epsilon$ is picked small just to avoid issues with continuity). Then this f(x) has period 3 as the set {2/9,4/9,8/9} is a periodic orbit of order 3, however there are intervals in the domain that are not chaotic.
>
> In general, this is also related to the reviewer's question on the observability of chaos. We believe it is true that if we can show Devanay chaos then this will imply observability, mainly due to transitivity. Two related works here that study the measure of chaos is [2,3].
>
> [1] Devaney’s chaos or 2-scattering implies Li–Yorke’s chaos by Wen Huang, Xiangdong Ye (Topology and its Applications 117.3 (2002): 259-272)\
> [2] On Scrambled Sets for Chaotic Functions by A. M. Bruckner and Thakyin Hu\
> [3] Positive Measure Scrambled Sets of Some Chaotic Functions by Ashvin Varada Rajan
>
> --- Hyperbolicity and Robustness: We thank the reviewer for this suggestion. We believe the reviewer is right that hyperbolicity should work for guaranteeing exploding gradients. To check this, we performed a set of experiments similar to those shown in Figure 3 of our paper, but this time we used the TANH activation, instead of a ReLU. With tanh activations, we also observed chaotic behavior as measured by the frequency of period 3 (last column of Fig. 3). Specifically, if instead of ReLU we used tanh activations the last column would be:
>
> (with TANH) Chaos Frequency %\
> 11\
> 9.7\
> 7.6\
> 2.6\
> 4.6\
>
> Moreover, we would like to point out that sometimes robustness of chaos to small perturbations is not guaranteed, as it may happen that smal modification of a function f(x) that has period 3, may lead to another function f'(x) that does not have period 3. A particular example is to take f(x) to be the triangle wave where the slope is $\phi=1.618...$, i.e., the golden ratio.  It is known [1] that any small $\epsilon$-perturbation that reduces the slope to $\phi-\epsilon$ will break period 3, so the resulting f'(x) does not contain a point of period 3. However, since the slope of f'(x) is still larger than 1, the exploding gradients are maintained and chaos survives only through higher-order periods like 5, 7, 9 etc.
>
> [1]: Better Depth-Width Trade-offs for Neural Networks through the lens of Dynamical Systems by Vaggos Chatziafratis, Sai Ganesh Nagarajan and Ioannis Panageas (ICML 2019)

---

> ### Author Response · Authors · 2022-08-02
> **Clarification on eigenvalues computation**
>
> We thank the reviewer for this comment. We believe the confusion comes from the different ways to determine the behavior of a dynamical system in the case that it is a continuous-time dynamical system vs the case that it is a discrete-time dynamical system.
>
> It is true that for continuous time dynamical systems $\dot{x}=Ax$ where the derivative is w.r.t. to time $t$ let's say, we should focus on the real part of the eigenvalues of $A$. However, for discrete-time dynamical systems, where $x_{t+1}=Ax_t$, we actually care about the magnitude of the largest eigenvalue of $A$, including its real and imaginary parts. In our case, since we have a discrete-time dynamical system w.r.t. to the number of iterations $t$ of the RNN, that's why we used the function "complexe_modulo" to determine stability/instability. For some references, see [1],[2] for example.
>
> Finally, we agree with the reviewer that finding more ways to evaluate the existence of chaos in higher dimensions, is a very interesting question for future research. From the literature on dynamical systems, we know that Li-Yorke chaos and Sharkovsky's theorem are not directly applicable to high dimensions. One approach could be to have a rigorous investigation of the Lyapunov exponents for different types of architectures as we vary several parameters (width,depth,activations,use of skip connections etc.).
>
> [1]: Stability of discrete dynamical systems by Maria Barbarossa (http:// m6auer.ma.tum.de/foswiki/pub/M6/Lehrstuhl/MatBio1_201011WS/discrete_dynamics.pdf)\
> [2]: https:// people.maths.bris.ac.uk/~maajh/ODEs/chap3.pdf

---

### Official Review · Reviewer_j5Sz · 2022-07-16

**Rating:** 4
**Confidence:** 3
**Soundness:** 2 fair
**Presentation:** 2 fair
**Contribution:** 1 poor

**Summary:**

This paper shows that randomly initialized RNNs can display a type of chaos called Li-Yorke chaos and the number of linear regions they have remains exponential with depth (unlike in feedforward nets).

**Questions:**

In experiments reported in Figure 3, do the authors use a standard RNN or an RNN with a clip operation as the one used for the proofs?

**Limitations:**

RNNs are not a very commonly used architecture in machine learning any more, so I doubt that these results will be of much interest to the NeurIPS audience. This paper in addition studies a peculiar type of RNN with a clip operation after the nonlinearity that calls into question even more the relevance of these findings for even standard RNN architectures that used to be used by the machine learning community.

It wasn’t really clear to me what the implications of Li-Yorke chaos or exponential number of linear regions are in practical terms beyond things like instability and exploding gradients etc. The results are in general very far removed from the interests and concerns of the overwhelming proportion of machine learning practitioners today. I believe a more specialized venue on dynamical systems may be more appropriate for this work.

Finally in Fig 4, having a Jacobian norm greater than 1 seems like a very weak proxy for what the paper is primarily concerned with, i.e. Li-York chaos.

**Strengths And Weaknesses:**

Strengths:

The work seems technically sound, although I have to admit I did not go through the proofs very carefully.

Weaknesses:

Please see questions and limitations below.

---

> ### Author Response · Authors · 2022-08-02
> **Question/Clarification for Figure 3**
>
> We thank the reviewer for the useful comments and questions.
>
> Regarding the experiments reported in Figure 3, the reported numbers were without the Clipping operation, so that the architecture is more faithful to practice. For completeness, we also performed additional experiments with the Clipping operation and the numbers are still ranging from 2%-9% for the probability of chaos. Specifically, following the setup of Fig. 3, out of 10000 runs we got: \
> (with CLIPPING) Chaos Frequency %\
> 9.7\
> 5.8\
> 5.7\
> 3.6\
> 2.2

---

> ### Author Response · Authors · 2022-08-02
> **Discussion on RNNs relevance**
>
> We thank the reviewer for the useful comments and feedback. Below we would like to address their concerns.
>
> The first comment is about the relevance of Recurrent Neural Networks (RNNs) in the machine learning community nowadays. We would like to address this comment in two ways, that we hope will help clarify our broader contribution.
>
> First of all, even if we suppose for a moment that RNNs were completely useless, our paper still proves in a rigorous way how small modifications in the architecture can have tremendous impact on the expressivity, trainability and chaotic behavior of neural networks. This sheds light to the role of the architecture for properties like exploding gradients, which is in general an impediment to learnability via gradient descent.
>
> To elaborate further on this first point, in our case, ``small modification’’ refers to the fact that RNNs use recurrent connections, whereas standard Feedforward Neural Nets (FNNs) do not. From prior works of Hanin and Rolnick [1],[2], the behavior (as far as expressivity/trainability is concerned) of FNNs under random initialization was well-understood. Our paper shows that RNNs exhibit completely different behavior and the culprit is indeed the recurrent links between activation units that force the same randomness across different iterations of the network. We show this by making a novel connection between RNNs and Li-Yorke chaos. For more details, please see lines 78-92 from our submitted paper.
>
> We view this result as a first step towards a principled approach for understanding other (very successful) architectures such as Residual Networks (ResNets) that use skip connections, in contrast to standard FNNs.
>
> Secondly, we also believe RNNs are an important family on their own sake and understanding their basic properties is of great interest. In general, RNNs are often employed for encoding temporal data or forecasting of future states. Their success is partly due to the internal memory architecture which allows these networks to better incorporate state information over the length of a given sequence. Because of their sequential nature, studying their stability (or instability) properties is of great interest. (e.g., see [4])
>
> Moreover, it is well-documented [5] that RNNs have multiple applications including: Machine translation, Robot control, Time series prediction, Speech recognition, Speech synthesis, Time series anomaly detection and many more. Important breakthroughs have taken place with the use of RNNs (e.g., [3]) and understanding the role of architecture that enables their success is crucial. In addition, while recurrent architectures are less prevalent for studying sequential language tasks, they remain applicable to simulating physical phenomena that can be represented with unknown or difficult to compute dynamical systems, like climate modeling. We hope that future work based on dynamical systems could elucidate basic properties in ResNets or other architectures that deviate from standard FNNs.
>
> [1]: Complexity of Linear Regions in Deep Networks by Boris Hanin, David Rolnick (ICML 2019)\
> [2]: Deep ReLU Networks Have Surprisingly Few Activation Patterns by Boris Hanin, David Rolnick (NeurIPS 2019)\
> [3]: Sequence to Sequence Learning with Neural Networks by Ilya Sutskever, Oriol Vinyals, Quoc V. Le (NeurIPS 2014)\
> [4]: Stable recurrent models by John Miller and Moritz Hardt (ICLR 2018)\
> [5]: "Recurrent neural network" entry of Wikipedia and list of Applications & References therein\
> [6] Using recurrent neural networks for localized weather prediction with combined use of public airport data and on-site measurements
>  by Jung Min Han, Yu Qian Ang, Ali Malkawi, Holly W.Samuelson (Building and Environment Journal, 2021)\
> [7] Solving differential equations with unknown constitutive relations as recurrent neural networks by Tobias Hagge, Panos Stinis, Enoch Yeung, Alexandre Tartakovsky (NeurIPS 2017)

---

> ### Author Response · Authors · 2022-08-02
> **Addressing Comment on Linear Regions and implications of chaos**
>
> We thank the reviewer for this question.
>
> In general, expressivity of a ReLU network is often captured by the number of its linear regions [1]. As a general rule, the more linear regions, the more expressive the network is. Also, the larger the width and depth, the more the linear regions. However, in surprising works of Hanin and Rolnick [2],[3], they showed that at initialization, deep Feedforward Neural Networks do not exhibit a number of linear regions close to the predicted upper bound. They also tied this with the problems of exploding gradients that govern the trainability of the system.
>
> However, their results only apply to FNNs, and so, a priori, recurrent models may exhibit a different behavior both in terms of linear regions, and in terms of their trainability. It has been well-known how learning long-term dependencies in recurrent networks is difficult because of the problem of vanishing and exploding gradients [4],[5] and this was actually our starting point that motivated our investigation. Our paper demonstrates how linear regions, trainability and exploding gradients are all intimately related; the underlying reason is the chaotic dynamics at initialization for the case of (a simple model of) RNNs. One implication of chaos is that with probability that does not depend on the network width, the randomly initialized RNN could exhibit very unpredictable behavior.
>
> For these reasons, we do not believe that the reviewer's phrase "The results are in general very far removed from the interests and concerns of the overwhelming proportion of machine learning practitioners today." is fair or accurate, given the large body of literature that studies expressivity, stability/instability, exploding/vanishing gradients at initialization and more.
>
> [1] On the Number of Linear Regions of Deep Neural Networks by Guido Montúfar, Razvan Pascanu, Kyunghyun Cho, Yoshua Bengio (NeurIPS 2014)\
> [2]: Complexity of Linear Regions in Deep Networks by Boris Hanin, David Rolnick (ICML 2019)\
> [3]: Deep ReLU Networks Have Surprisingly Few Activation Patterns by Boris Hanin, David Rolnick (NeurIPS 2019)\
> [4]: A Simple Way to Initialize Recurrent Networks of Rectified Linear Units by Quoc V. Le, Navdeep Jaitly, Geoffrey E. Hinton (NeurIPS 2015)\
> [5] Stable recurrent models by John Miller and Moritz Hardt (ICLR 2018)\

---

> ### Author Response · Authors · 2022-08-02
> **Figure 4 Clarification**
>
> We thank the reviewer for this comment. For the case where Li-Yorke chaos can be verified, i.e., the one-dimensional dynamical system, we also have the probability of chaos as measured by the frequency of period 3 in the output (see Fig.3)
>
> Regarding higher dimensions, as we state in the paper (line 313) since high dimensions do not have clean characterizations of chaos such as the existence of a periodic 3 point in the domain, we instead detect chaos by relying on a necessary condition: Jacobian’s spectral norm being more than 1.
>
> Finding more ways to evaluate the existence of chaos in higher dimensions, we believe this to be an interesting question for future research. From the literature on dynamical systems, we know that Li-Yorke chaos and Sharkovsky's theorem are not directly applicable to high dimensions. One approach could be (as per reviewer's "EXTU" suggestion) to have a rigorous investigation of the Lyapunov exponents for different types of architectures as we vary several parameters (width,depth,activations,use of skip connections etc.).
>
> We would like to point out here however, that we performed experiments with other types of activations units and higher values for the width and the depth and we observe that these variants too exhibit period 3. For example, with tanh activations as per another reviewer's suggestion, we also observed chaotic behavior as measured by the frequency of period 3 (last column of Fig. 3). Specifically, if instead of ReLU we used tanh activations the last column would be:
>
> (with TANH) Chaos Frequency %\
> 11\
> 9.7\
> 7.6\
> 2.6\
> 4.6

---

### Author Response · Authors · 2022-08-07
**Looking forward for reviewers' feedback after rebuttal?**

We thank the reviewers' for their comments and effort.

We look forward for your feedback after our responses. Let us know if there is anything more that is needed.

---

### Meta-Review · Area_Chair_MLQC · 2022-08-27

**Recommendation:** Accept
**Confidence:** Less certain

**Metareview:**

This submission is borderline.  Reviewer EXTU praises its theoretical contribution and highly recommends acceptance.  Reviewers niom and YhQb are more tepid but still support acceptance in light of the sound theoretical analysis.  Reviewer j5Sz acknowledges that the analysis is sound, but believes the models to which it pertains (RNNs, with certain unconventional features) are of little interest to the community, and therefore argues for rejection.  While I agree with j5Sz that the paper may not be of immediate interest to many practitioners, I believe the theory it delivers is worthy on its own, and may lead to further theoretical developments that will apply to more contemporary neural networks.  I thus recommend acceptance.

**Award:**

No

---

### Decision · Program_Chairs · 2022-09-14

Accept